# Genome-Based Analysis of Virulence Factors and Biofilm Formation in Novel *P. aeruginosa* Strains Isolated from Household Appliances

**DOI:** 10.3390/microorganisms10122508

**Published:** 2022-12-19

**Authors:** Annika Kiel, Ines Creutz, Christian Rückert, Bernhard Peter Kaltschmidt, Andreas Hütten, Karsten Niehaus, Tobias Busche, Barbara Kaltschmidt, Christian Kaltschmidt

**Affiliations:** 1Department of Cell Biology, Faculty of Biology, Bielefeld University, 33615 Bielefeld, Germany; 2Proteome and Metabolome Research, Faculty of Biology, Bielefeld University, 33615 Bielefeld, Germany; 3Center for Biotechnology (CeBiTec), Bielefeld University, 33615 Bielefeld, Germany; 4Department of Thin Films and Physics of Nanostructures, Center of Spinelectronic Materials and Devices, Faculty of Physics, Bielefeld University, 33615 Bielefeld, Germany

**Keywords:** *Pseudomonas aeruginosa*, biofilm, virulence factors, household appliances, whole genome sequencing, prophages

## Abstract

In household washing machines, opportunistic pathogens such as *Pseudomonas aeruginosa* are present, which represent the household as a possible reservoir for clinical pathogens. Here, four novel *P. aeruginosa* strains, isolated from different sites of household appliances, were investigated regarding their biofilm formation. Only two isolates showed strong surface-adhered biofilm formation. In consequence of these phenotypic differences, we performed whole genome sequencing using Oxford Nanopore Technology together with Illumina MiSeq. Whole genome data were screened for the prevalence of 285 virulence- and biofilm-associated genes as well as for prophages. Linking biofilm phenotypes and parallelly appearing gene compositions, we assume a relevancy of the *las* quorum sensing system and the phage-encoded bacteriophage control infection gene *bci*, which was found on integrated phi297 DNA in all biofilm-forming isolates. Additionally, only the isolates revealing strong biofilm formation harbored the ϕCTX-like prophage Dobby, implicating a role of this prophage on biofilm formation. Investigations on clinically relevant pathogens within household appliances emphasize their adaptability to harsh environments, with high concentrations of detergents, providing greater insights into pathogenicity and underlying mechanisms. This in turn opens the possibility to map and characterize potentially relevant strains even before they appear as pathogens in society.

## 1. Introduction

*Pseudomonas aeruginosa* has a diverse and dynamic genetic composition that facilitates the colonization of various environments, including household appliances such as washing machines [1,2,3]. We and others have analyzed various bacteria derived from washing machines [2,3,4,5]. We discovered that all bacteria derived from natural habitats, such as soil and water, were selectively enriched in washing machines that acted as bioreactors. These bacteria have the capability to adapt to environments with harsh conditions, making these opportunistic bacterial pathogens a challenging threat [6,7]. In particular, *P. aeruginosa* is well known for its strong biofilm-producing potential, making it undesirable both in household appliances and clinics [8,9,10]. In the clinical daily routine, *P. aeruginosa* is particularly associated with chronic lung infections in patients suffering from cystic fibrosis (CF). The evolution driving *P. aeruginosa* in CF infections has been extensively studied [11,12,13,14]. An important focus of these studies is the understanding of the genetic composition responsible for successful infections and antibiotic resistance. A hallmark of the *P. aeruginosa* infection of the lungs in patients with CF is the ability to form biofilms [15,16].

There are two types of *P. aeruginosa* growth: the free-living motile planktonic type or the sessile/biofilm type, which is enclosed in an extracellular polymeric substance (EPS), adhering to each other and/or surfaces or interfaces [17]. While the motile life form of *P. aeruginosa* has a more virulent phenotype, switching to the sessile form of life confers survival advantages [18]. *P. aeruginosa* uses flagella and pili for locomotion, with Type VI pili being associated with virulent properties [11]. Swimming bacteria can be identified by the host immune system, while flagella or other mobility components of the pathogen are recognized and induce signaling pathways and inflammatory responses [19]. During initial attachment, bacteria lose their motility and begin to form a monolayer. Subsequently, the formation of network-like structures, such as the extracellular matrix, takes place. When maturing, continuous microcolonies (multilayer) are formed which can be embedded in EPS, consisting of polysaccharides, proteins, nucleic acids and lipids [20,21]. The highly structured protective EPS matrix is a dynamic ecosystem that can be continuously remodeled, which enables bacteria to adapt to environmental changes [20,22,23,24]. Within biofilms, bacteria reveal higher tolerance against antibiotics, disinfectants and detergents [25,26,27]. Bacteria within biofilms profit from advantages over their mobile free-living counterparts [18]. Biofilms in household appliances differ in composition depending on the sampling location, revealing a strong sensitivity towards environmental factors such as detergent concentrations, as well as material and surface properties [5]. Other surface structures and important virulence factors of *P. aeruginosa* are the outer membrane component secretion systems (Types 1, 2, 3, 5 and 6) [28]. They secrete various factors such as toxins and proteins related to biofilm formation and adhesion (Type 5 and Type 6 secretion systems) [29,30,31,32,33]. The Type 3 secretion system, as one of the most important virulence factors of *P. aeruginosa*, secretes the four well-known effectors ExoS, ExoT, ExoU and ExoY, which play crucial roles during infections [34,35,36,37]. Virulence factor production, as well as modulating of biofilm production, is regulated by the bacterial cell-to-cell communication system called quorum sensing [28,38]. Therefore, numerous *P. aeruginosa* virulence factors such as flagella, pilli, secreted toxins, etc., are directly linked with biofilm formation, making them interesting targets for biofilm research.

For better characterization of the molecular biology of biofilm formation, we have sequenced the genomes of four *P. aeruginosa* washing machine (WM) isolates. The availability of whole genome sequencing technologies and genomic data has not only led to a better understanding of infections and virulence genes but also revealed that a significant portion of the bacterial genome is of viral origin [39,40]. Some components of this viral DNA contain intact, functional prophages, while other parts consist of prophage-like elements. Cryptic, questionable or incomplete prophages, genomic islands and plasmids might have been obtained by horizontal gene transfer [39,41]. It has been reported that this viral genetic material has an influence on virulence and also on the biofilm formation of the bacterial host [39,42,43,44,45,46]. Genes such as cholera toxin or Shiga toxin can be carried which directly influence the virulence and the metabolic activities. In particular, spontaneous prophage induction, triggered, for example, by the SOS response or by stochasticity in bacterial host gene expression, plays a crucial role in the activation of the genetic elements of viral origin [35]. For instance, Shiga toxin (Stx) release in the gut by a small number of Shiga toxin-producing *E. coli* (STEC) bacteria causes diarrhea and is mediated by the spontaneous induction of Stx-encoding prophages. This enables the spread of uninduced lysogenic *E. coli* bacteria within the gut and in consequence increases the fitness of the bacterial population [46]. *P. aeruginosa* biofilm development is highly influenced by Pf filamentous prophages [43]. For example, cell lysis induced by the Pf4 prophage induction releases eDNA which is an important component of the extracellular matrix and therefore promotes *P. aeruginosa* biofilm formation [39,47]. During late stages of biofilm formation, the activation of Pf4 via elevated ROS levels also enhances the dispersal of the biofilm [48,49]. In this study, we aim to gain an understanding of the importance of specific virulence factors and prophage integration in *P. aeruginosa* strains isolated from household appliances and their role in biofilm formation. We hypothesize that bacterial strains found in environments such as the washing machine have an increased biofilm formation potential, which is encoded within the genome.

While most studies focus on strains isolated from infected patients, the study presented here focuses on environmental isolates. As presented in our previous study [4], twenty-nine bacterial isolates could be identified from different parts of household washing machines. Bacterial isolates were tested for their biofilm-forming potential. *P. aeruginosa* was found to be best in biofilm production on various material surfaces. Strong biofilm-forming potential could be determined in two of the four sequenced isolates. General genomic features, evolutionary relationships, prophage regions as well as virulence and biofilm-related genes were examined. The strong biofilm-producing isolates were the only ones harboring the ϕCTX-like *Pseudomonas aeruginosa* phage Dobby and an intact copy of the bacteriophage control infection (*bci*) gene. These novel data imply that prophages play a crucial role in the biofilm formation of wild-type *P. aeruginosa* isolates.

## 2. Materials and Methods

### 2.1. Bacterial Strains

Four *Pseudomonas aeruginosa* isolates, isolated from domestic washing machines [4], were investigated. Isolation sites were the following: B1.2 isolated from the detergent compartment, B2.1 isolated from the detergent enema, C1.3 isolated from the detergent compartment and C4.2 isolated from the door sealing rubber (elastomers, rubber). The bacterial isolates were inoculated from frozen stocks stored at −80 °C and cultured on LB agar plates for 24 h prior to experimental use.

### 2.2. Phenotypic Investigations

#### 2.2.1. Cultivation

For all experiments, the four *P. aeruginosa* isolates were inoculated from fresh LB agar plates and pre-cultured in 10 mL LB medium at 37 °C overnight with agitation at 120 rpm without additional lightning. 

The growth behavior of the four *P. aeruginosa* WM isolates was investigated in 5 mL LB medium without agitation at 37 °C. Pictures of the cultures were taken after 24 h, 48 h and 72 h. 

#### 2.2.2. Microtiter Plate Assay (MTP)

To determine the biofilm formation potential of the four *P. aeruginosa* isolates, the quantitative microtiter plate assay described by Christensen et al. with modifications was performed [50]. LB, TSB and TSB supplemented with glucose (1%) medium were used. For the experiment, *P. aeruginosa* stationary phase overnight cultures were adjusted to OD600 = 0.01 (approximately 5 × 10^6^ CFU/mL). Polysterol 96 flat bottom well plates (SARSTEDT AG & Co. KG, Nümbrecht, Germany) were used for the experiment. For each isolate, 6 wells were filled with 200 µL of each medium inoculated with *P. aeruginosa* (6 technical replicates). Due to the frequent temperature-induced edge effect, the outer wells were not used. Plates were incubated for 24 h, 48 h and 72 h hours at 37 °C under static conditions. After incubation, the planktonic cells were washed away by filling the wells three times with 200 µL physiological saline (0.9% NaCl). Afterwards, plates were air-dried for 24 h. Remaining biofilm residues were stained with 0.1% crystal violet (CV) for 30 min. Crystal violet was discarded and wells were washed three times with bidest H_2_O. The remaining stained biofilm was dissolved with 200 µL of 30% acetic acid. A 2-fold dilution was performed in a fresh 96 flat bottom well plate with an end volume of 200 µL. Optical densities were spectrophotometrically measured at 590 nm using a PowerWave microplate reader (BioTek, Winooski, VT, USA).

Additionally, the influence of washing detergents for biofilm detachment and production was tested exemplarily for the strong biofilm-producing B2.1 *P. aeruginosa* isolate. Due to dissolving issues of powder detergents, liquid washing detergents (Henkel, Düsseldorf, Germany) were chosen: detergent 1 = Persil Color Kraft-Gel, detergent 2 = Persil Universal Kraft-Gel and detergent 3 = Perwoll wool and fine detergent. To test the detachment of biofilms by detergents, biofilms were grown for 24 h in 96 well plates as described above using LB medium. The inoculum was adjusted to OD600 = 0.01. After the incubation period of 24 h, the biofilm was washed two times with bidest H_2_O. Afterward, the wells were filled with the detergent solutions in different concentrations. Liquid detergents were dissolved in bidest H_2_O at concentrations of 10, 7.5, 5, 0.5, 0.05, 0.005, 0.0005 and 0.00005 mL/L. For each tested washing detergent concentration, 4 technical replicates (wells) were performed. Plates were incubated for 30 min at 30 °C. The washing detergent solution was discarded and wells were washed three times with bidest H_2_O. Plates were air-dried overnight and stained with crystal violet as described above. 

To test biofilm formation in the presence of a liquid detergent, the detergent was added directly to the culture LB medium. The same detergent concentrations as described above were used. Optical densities were adjusted to OD600 = 0.01 and plates were incubated for 24 h at 37 °C under static conditions. Afterward, the medium was discarded and the plates were washed three times and air-dried overnight. Crystal violet staining was performed as described above. *P. aeruginosa* incubated in LB medium without washing detergents served as a control in both experimental setups.

#### 2.2.3. SEM of Attached Cells and Biofilm Formation

For an ultrastructural observation of attached cells and biofilm formation, the four isolates were grown in 6 well plates (SARSTEDT AG & Co. KG, Nümbrecht, Germany) containing glass slides with a size of 22 mm× 22 mm × 0.4 mm (Assistent, Karl Hecht GmbH% Co Kg, Sondheim vor der Rhön, Germany). For each isolate, 2 mL bacterial suspension (in LB-medium), adjusted at OD600 = 0.1, was poured into the well. The four *P. aeruginosa* isolates were incubated for 24 h at 37 °C under static conditions. After the incubation period, planktonic cells were washed away by submerging the glass slide two times in physiological saline (0.9% NaCl) followed by one time in bidest H_2_O. The remaining attached cells/biofilm was fixed with half-strength Karnovsky’s solution (2% paraformaldehyde, 2.5% glutaraldehyde) for 30 min. The fixed samples were dehydrated using 50, 70, 80, 90, 95 and 100% (*v*/*v*) graded ethanol followed by t-butyl alcohol. The SEM used was a Helios NanoLab DualBeam 600 (FEI Company, Hillsboro, OR, USA) scanning electron microscope. Prior to analysis, the samples were sputter coated with a layer of 4 nm ruthenium and connected to the stage with conducting tape to ensure proper conduction. The measurements were carried out at an acceleration voltage of 5 kV with a beam current of 0.17 nA.

### 2.3. Whole Genome Data Analysis

#### 2.3.1. Whole Genome Sequencing and Genome Assembly

Genomic DNA was extracted from overnight cultures. The NucleoSpin^®^ Microbial DNA Extraction Kit (Macherey-Nagel, Düren, Germany) was used following the manufacturer’s guidelines. 

#### 2.3.2. Nanopore Library Preparation and GridION Sequencing

A sequencing library with genomic DNA was prepared using the Nanopore DNA Ligation Sequencing Kit (SQK-LSK109, Oxford Nanopore Technologies, Oxford, UK) according to the manufacturer’s instructions. Sequencing was performed on an Oxford Nanopore GridION Mk1 sequencer using an R9.4.1 flow cell, which was prepared according to the manufacturer’s instructions. Basecalling was performed using guppy v5.0.16 with the super-accurate basecalling model.

#### 2.3.3. Illumina Library Preparation and MiSeq Sequencing

Whole genome shotgun PCR-free libraries were constructed from 5 μg of gDNA with the TruSeq DNA PCR-Free Sample Preparation Kit (Illumina, San Diego, CA, USA) according to the manufacturer’s protocol. Quality of the resulting libraries was controlled by using an Agilent 2000 Bioanalyzer with an Agilent High Sensitivity DNA Kit (Agilent Technologies, Santa Clara, CA, USA) for fragment sizes of 500–1000 bp. Paired-end sequencing was performed on the Illumina MiSeq platform (2 × 300 bp, v3 chemistry).

#### 2.3.4. Assembly and Annotation

All sequencing data were checked with FastQC prior to assembly [51]. The assemblies were performed using flye v2.9 [52] using standard parameters and --nano-hq for data input. The assembly graphs were checked (and in case of C1.3 and C4.2 resolved) with Bandage [53], resulting in one contig per replicon. The resulting contigs were polished first with medaka v1.6.0 using standard parameters and model r941_min_sup_g507 and afterwards with Illumina short-read data using Pilon v1.22 (--fix all --changes --frags <Sorted BAM file>) run in iterative cycles using Bowtie2 v2.3.2 for mapping (-X 750 --no-unal) of the reads in paired mode. The assembled genomes were annotated using the NCBI PGAP pipeline [54,55]. 

#### 2.3.5. Data Availability

All raw sequence data as well as the assembled genomes are available via the BioProjects PRJNA838003 (B1.2), PRJNA838002 (B2.1), PRJNA838005 (C1.3) and PRJNA838004 (C4.2), respectively.

#### 2.3.6. Phylogenetics, Pan-Genome, Typing and Prophage Investigation

For an additional identification of our four isolates, the genome-based classification and identification by the Type (Strain) Genome Server (TYGS) provided by the Deutsche Sammlung von Mikroorganismen und Zellkulturen (DSMZ) was performed [56,57]. A genome-scale GBDP tree and a 16S rRNA gene sequence tree were provided. Additionally, a phylogenetic tree comparing the four WM isolates to 233 complete *P. aeruginosa* genome sequences, including the popular/reference strains PAO1 (NC_002516.2), LESB58 (NC_011770.1), UCBPP-PA14 (NC_008463.1) and PA7 (NC_009656.1), was built. Whole genome sequences were downloaded from the *Pseudomonas* Genome Database (PGDB) [58,59]. Pairwise average nucleotide identities (ANIs) were calculated using ORTHOANIU [60], with USEARCH [61]. From the pairwise ANI values, a distance matrix was created which was subsequently used to construct a neighbor-joining tree [62] using MEGA X [63]. The tree was visualized using the online tool Interactive Tree of Life (ITOL) [64]. The pan-genome was analyzed and visualized using EDGAR 3.0 [65,66]. Multilocus sequence typing (MLST) was performed by using the pubMLST database [67] to investigate the sequence type (ST) of the four *P. aeruginosa* isolates. Additionally, the *P. aeruginosa* serotype was examined by using the *P. aeruginosa* serotyper PAst 1.0 [68,69]. The identification of prophages was carried out using the web tool PHASTER [70].

#### 2.3.7. Biofilm-Associated and Virulence Genes

The four *P. aeruginosa* genomes were screened for the presence and absence of virulence- and biofilm-associated genes. A set of 285 virulence- and biofilm-associated genes of the reference strain PAO1 were compiled from the Virulence Factor Database (VFDB) [71]. Additional genes associated with biofilm formation and quorum sensing (*pel, psl* and quorum sensing systems *MvfR* (also known as *pqsR), IqsR*) were chosen from the literature [72,73]. The BLAST Ring Image Generator (BRIG) [74] was used to visualize BLAST search matches between the reference genome and the studied WM isolates. BRIG was used with default BLAST+ settings with task as BLASTn without any additional filtering. A list of all virulence- and biofilm-associated genes can be found in Appendix A. The virulence gene *exoU* was used from the reference strain *Pseudomonas aeruginosa* UCBPP-PA14 because it is not present in PAO1.

#### 2.3.8. Statistical Analysis

For statistical analysis, GraphPad Prism V8.3.0 software (GraphPad Software, Inc., San Diego, CA, USA) was used. Test for normality was conducted using D’Agnostino and Pearson normality tests. To evaluate differences between multiple groups, we performed unpaired Brown–Forsythe and Welch ANOVA tests. For data sets where no normal distribution can be assumed, a Kruskal–Wallis test was performed. A significance value of *p* ≤ 0.05 was considered as statistically significant. The data are presented as means ± standard error of the mean (SEM).

## 3. Results

In a previously published study, the bacterial spectrum found in washing machines was investigated [4]. Six washing machines were tested and different material surfaces were swabbed including the detergent compartment, detergent enema, washing drum, sight glass and door sealing rubber. In a previous study, 29 bacterial isolates could be clearly identified by MALDI-TOF-MS. *P. aeruginosa* could be identified in two washing machines, in three of the five compartments (Table 1) and was proven to be the strongest biofilm producer. In the current study, the differences between these four isolates in terms of biofilm formation were further investigated, phenotypically and genetically.

### 3.1. Bacterial Growth, Attachment and Biofilm Formation

First insights into the different phenotypes of the four *P. aeruginosa* isolates were made. Cultivation of *P. aeruginosa* isolates in LB medium revealed that after an incubation period of 72 h, only the C1.3 isolate produced a yellow-greenish color at the water–air interface (see Figure 1A). Strong pellicle formation at the liquid–air interface was visible for all strains except for the C4.2 isolates. We further investigated the biofilm formation potential by the microtiter plate assay. After an incubation period of 24 h, strong biofilm formation could be measured for the two B isolates in all tested media. The B1.2 isolate produced significantly higher amounts of biofilm compared to the C strains. The strongest biofilm formation could be observed in TSB supplemented with glucose as an additional carbon source. The B strains also showed the strongest measurable biofilm after 48 h and after 72 h (see Appendix A).

To gain insights into the biofilm nanostructure, ultrastructural analyses were performed by using SEM. The four isolates were incubated for 24 h on glass slides with LB medium. Horizontal growth and typical biofilm structures such as supply channels and mucus production could be observed in the two B strains only. However, for the isolates C1.3 and C4.2, network-like structures with inner connecting filaments and initial mucus production were also observed. This indicates that these strains are capable of biofilm formation but not as quickly and efficiently as the B isolates.

Since the best biofilm-producing strains were found in areas with high concentrations of detergents, biofilms were additionally investigated in terms of production and detachment in the presence of detergents (Figure 2). One of the strong biofilm-producing isolates (B2.1) was used for these experiments. Already-formed biofilms show a high tolerance against liquid detergents. The biofilms could not be detached even with high concentrations of detergents (Figure 1A). An increased optical density can even be observed for detergent 3 and for detergent 2 at low concentrations. This indicates that the detergents in subinhibitory concentrations either stimulate the biofilm growth very quickly or detergent components adhere to the biofilm. A slight detaching effect can only be observed with detergent 1 at a concentration of 2.5 mL/L. On the other hand, biofilm formation in the presence of detergents with concentrations above 0.05 mL/L is decreased up to 50% compared to the control (Figure 2B). Interestingly, a slightly increased effect of biofilm formation can be observed at a concentration for detergent 1 at 0.0005 mL/L and for detergent 2 at 0.005 mL/L. Taken together, these results indicate that already-formed biofilm cannot be removed by liquid detergents. It can be speculated whether liquid detergents even have a biofilm growth-promoting effect. However, in the presence of higher concentrations of detergents, decreased biofilm formation can be observed. 

### 3.2. Genomic and Phylogenetic Features of P. aeruginosa WM Isolates

Whole genome sequencing was performed to gain insights into the genomic differences which might explain the observed phenotypic differences. In addition to the MALDI-TOF-MS typing, we identified the species and type (strain), by the use of genomic data. Draft genomes were used for a genome-based taxonomy using the Type (Strain) Genome Server (TYGS), provided by the DSMZ [56,57]. Phylogenetic trees at the genome level were inferred from the whole genome sequences using BLAST Distance Phylogeny (GBDP) (Figure 3) and also based on the 16S rRNA sequences (see Appendix A). Both phylogenetic investigations reveal that the four WM isolates are of the species *P. aeruginosa*. However, the C1.3 isolate has a slightly different GC content of only 65% compared to that of the average reference genomes of 66% (Table 2).

Additionally, genomic features were investigated for all isolates and compared with four reference strains (Table 2). The genome length of the four isolates differed between the B and C strains. Compared to the mean genome length of the reference strains (mean length = 6.498.037 bp), the genomes of both C isolates are substantially larger (about 1 Mbp). The genome length of the B isolates is about 110,000 bp smaller in size compared to the reference genomes. For the B isolates, the total length of the genomes differs only by 14 bp. The absolute numbers of genes as well as the coding sequence (CDS) are about 1000 genes more in the C isolates than in the B isolates. The C isolates also harbor more genes compared to the reference strains. Both B isolates have the same number of total genes as well as CDS. The total number of genes is in range with the mean number of genes of the reference strains (mean of total genes = 6022). In silico multilocus sequence typing (MLST) and PAst serotyping revealed the same sequence (ST369) and serotype (O6) for both B isolates. The C1.3 isolate has the ST313 and belongs to the serotype group O1. C4.2 reveals the sequence type ST2844 and belongs to the serotype O13. In addition, the pan-genomes of the four strains were examined (Figure 4). The core genome of the four strains consists of 5325 CDS. The C isolates have a high number of singleton genes (singleton genes C1.3 = 1037, singleton genes C4.2 = 819). Interestingly the B isolates harbor nearly no singleton genes (singleton genes B1.2 = 1, singleton genes B2.1 = 0). The B isolates shared 222 orthologous CDS exclusively.Taken together, the results of the general genomic features indicate that the B isolates might belong to the same strain. Clear differences between the B and C strains can also be seen at the genomic level in addition to the phenotypic analyses.

Additionally to the genome-scale GBDP tree, a phylogenetic analysis of the four *P. aeruginosa* WM isolates and 233 complete genome sequences downloaded from the Pseudomonas Database was built (Figure 5). The 233 genome sequences taken from the Pseudomonas Database represent a global *P. aeruginosa* collection. The analyzed strains cluster into two major groups (Figure 5, colored red and blue), with some strains including PA7 as a taxonomic outlier (Figure 5, colored in yellow). This observation is consistent with several other studies on the phylogenesis of *P. aeruginosa* [75,76,77,78]. Phylogenetically, both B and the C4.2 isolates are localized in the same phylogenetic cluster (Figure 5 colored in red). This cluster also includes the widely studied reference strain PAO1 [1] and the notable CF strain LESB58 [79]. The C4.2 isolate shows a close relation to the *P. aeruginosa* isolate H47921 isolated from cancer patients in the USA [80]. Both B isolates occur in their own phylogenetic subgroup. The C1.3 isolate is located in the blue cluster, which also includes the popular virulent strain PA14 [81]. C1.3 clusters together with the multidrug resistantBAMCPA07-48, isolated from a combat injury wound in the USA [82]. These results show that two of our washing machine isolates are closely related to two known infectious *P. aeruginosa* strains.

### 3.3. Identification of Prophages

In the case of strains C1.3 and C4.2, we observed an incomplete assembly of the chromosome despite the use of long reads with a read N50 of 7.7 kb (C1.3) and 13.7 kb (C4.2). In both cases, three contigs were assembled, and in both cases, one of these contigs displayed signs such as an increased coverage, a G + C content diverging from the rest of the chromosome and a size (23.8 kb or 36.5 kb) that is often found in prophages that are present in either several copies and/or a mixed state between prophage and episome. To test this hypothesis, we therefore performed a phage screening (using PHASTER). All isolates reveal multiple prophage regions (Figure 6). For the C1.3 isolate, no complete assembly was possible and therefore three parts of the whole genome have been analyzed. A list of all intact, questionable and incomplete integrated prophages is provided in the Appendix A. The B isolates reveal two intact prophage regions, the Pseudo_phi297 and the Pseudo_Dobby, which appear at the exact same positions. This is another indication that the two B isolates are probably two subpopulations of one strain which has colonized two different locations of a washing machine. The C1.3 isolate has four intact prophage regions and the C4.2 isolate has ten intact prophage regions. Both C strains harbor the Pseudo_YMC11/02/R656 prophage. The C4.2 isolate also has the Pseudo_phi297 prophage integrated. The prophage phi297 has been reported to be homologous to the AUS531phi phage [42]. The bacteriophage control infection (*bci*) gene encoded on the AUS531phi phage showed an inducing effect on biofilm formation. The presence of the *bci* gene within the three phi297 prophage regions found here was investigated using BLAST (see Appendix A). For the B isolates, the complete gene could be identified with a query cover of 95%. The phi297 prophage within the C4.2 isolate revealed a lack of 10 base pairs at the end of the sequence including the stop codon, nevertheless revealing a query cover of 97%. In addition to the genomic comparison, the influence of the mutations was tested on the codon level using the multiple alignment tool Clustal Omega (see Appendix A). For both phi297 phages within the B isolates, three amino acids (AS) differ in comparison to the *bci* protein sequence obtained from the AUS531phi phage. The phi297 prophage within the C4.2 isolate differs only in one AS, but is missing the last four AS including the stop codon. 

Only the strong biofilm-producing B isolates harbor the Pseudo_Dobby prophage. Additionally to the prophage Dobby, both B isolates contain three questionable copies of the Pseudo_Pf1 prophage. The C1.3 isolate also harbors one questionable copy of the Pf1 prophage region. The identification and characterization of prophage regions within the WM isolates may indicate that the prophage Dobby has a biofilm-inducing effect, as this phage was only present in the strong biofilm-producing B isolates.

### 3.4. Virulence- and Biofilm-Associated Genes

Based on the Virulence Factors of Pathogenic Bacteria Database (VFDB) and literature, virulence genes associated with cystic fibrosis (CF) and biofilm formation were selected and screened in the WM isolates [71,72,73]. A total of 285 virulence genes of the reference strain PAO1 were compared using BLAST and BLAST Ring Image Generator (BRIG) for visualization (Figure 7). Genes associated with adherence, mobility, antimicrobial activity/competitive advantages, alginate, quorum sensing, effector delivery system, exoenzymes, exotoxins, immune modulation, nutritional metabolic factors and regulation were chosen for comparison. Additionally to the 285 virulence genes, the prevalence of the *pel* and *psl* polysaccharide locus, the *psl* and pyoverdine (iron-binding peptide) operon regulator *ppyr* and the quorum sensing regulators *mvfR* (also known as *pqsR,* including the transcription factors *pqsABCDE*) and the transcription factors belonging to the IqsR system *ambBCDE* were investigated.

The complete presence of the nutritional/metabolic factor pyoverdine locus (*fpvA*, *pvdADEFGHIJLMNOPQSY*) can only be observed in the C1.3 isolate. The other three tested strains show only a similarity of about 70% to most of the genes within the locus of PAO1. The genes *pvdA, pvdD, pvdE, pvdF, pvdI, pvdK, pvdP* and *pcdY* are uniquely present in the C1.3 isolate. The yellow-greenish compound pyoverdine was phenotypically observed only in the C1.3 isolate in stationary phase cultures cultivated under static conditions. The effector *pldA*, secreted by the Type VI secretion system, is again only present in C1.3. The effector exoS secreted by the Type III SS shows only a similarity of about 70% in the C1.3 isolate while, on the other hand, the effector *exoU* is present only in this isolate. Interestingly, the genes *lasI* and *lasR* associated with quorum sensing are not present in the C1.3 isolate. Lipopolysaccharide (LPS) genes associated with immune modulation (*hisF2, hisH2, PA3142, PA143, PA57, wbpABDEGHIJKLM* and *wzxyz*) were not present in any of the tested isolates. Only LPS genes *waaA, waaC, waaF, waaG* and *waaP* could be detected in all isolates. Genes associated with mobility and flagella also revealed an interesting pattern. The genes *fleI/flag, fleP, flgK, flgL, fliC, fliD* and *fliS* as well as the Type IV pili genes *pilA, pilC* and *pilB* are not present in any of the isolates. The presence of the *pel* and *psl* polysaccharide genes, the quorum sensing system multiple virulence factor regulator (*mvfR*, also known as *pqsR*) including *pqsABCD* and the transcription factors *ambBCDE* belonging to the *IqsR* quorum sensing system could be found in all isolates (see Appendix A). 

## 4. Discussion

Due to the clinical relevance of *P. aeruginosa*, the majority of published research investigates virulence factors in isolates from patients suffering of acute and chronic infections. However, strains isolated from household appliances can constitute the reservoir for clinical (outbreak) strains [2,3,4,83,84]. Consequently, shedding light on the virulence of household isolates of *P. aeruginosa* could in turn provide a better insight into pathogenicity and underlying mechanisms, thus enabling an understanding of the different phenotypes and genotypes present in the environment. This study looks a step ahead and concentrates on household-acquired *P. aeruginosa* isolates. Here, we have sequenced four WM isolates of *Pseudomonas aeruginosa* strains. The four strains significantly differed in biofilm formation. A strong biofilm formation could be observed for the B strains. Both B strains were isolated from parts of the washing machine which are in contact with high concentrations of washing detergents. Investigations of sensitivity against common commercially available washing detergents revealed high tolerance of B2.1 biofilms against the tested detergents. The B strains and C4.2 are relatively similar in genomic sequence whereas C1.3 is very different. Genomic analysis showed the presence of these strains (B1.2, B2.1 and C4.2) in a phylogenetic cluster. The C1.3 strain is more closely related to the highly virulent UCBPP-PA14 strain [85]. Both B strains differ in 221 genes compared to the C strains, but they are very similar in sequence to each other. In contrast, the C strains are very different in genomic structure and contain about 1000 genes each which were not present in the B strains. The four strains were isolated from two different washing machines. Thus, both phenotypic and genotypic differences may result from the different washing behavior of the users and the resulting selection pressure, but also from the different bacterial load within the washing machines. Strains that have a higher tolerance against external threats such as detergents are able to survive and are therefore found more frequently. However, many other environmental parameters may also be causative for these differences. Interestingly, genome comparison with 233 other fully sequenced strains identified closely related strains for the two C isolates, but none for the B isolates. Both strains showing phylogenetic similarity to the C isolates were originally isolated from patients in the USA. Therefore, the origin and the environmental influences resulting from the isolation source could also have an influence on the genotype and resulting phenotype.

Even though *P. aeruginosa* isolate C4.2 has all the genomic features indicative of biofilm formation like the B strains, there is no strong biofilm formation observable. One hypothesis in the literature is the activation of prophages inducing biofilm formation [43,45]. All WM isolates sequenced here reveal the integration of prophage regions within the genome. The B isolates reveal two, the C1.3 isolate four and the C4.2 isolate ten intact prophage regions. Additionally, all isolates contain incomplete prophage regions, which might occur due to incomplete excision events [39].

Prophages have been reported to play crucial roles in *Pseudomonas* virulence, chronic infections, antibiotic resistance and biofilm formation [44,86]. It is reported that Pf phage-producing *P. aeruginosa* bacteria within biofilms are more adherent to surfaces compared to bacteria that do not produce the Pf phage [43,49,86,87]. During biofilm formation, Pf virions accumulate within the biofilm matrix, giving the biofilm matrix liquid crystalline properties [87,88]. Bacteria within these crystalline biofilm matrices revealed higher resistance to desiccation [44]. This ability could be beneficial for biofilms that form in washing machines, since washing machines are often not used for several days, which leads to the drying out of biofilms. Another discussed mechanism for the increased formation of biofilms after prophage induction is the release of eDNA [43,89]. In contrast to previous studies on the impact of Pf phage, the WM isolates we describe here have only incomplete copies of Pf1 prophage. The B isolates reveal three questionable copies of the Pf1 phage and the C1.3 isolate harbors one copy. Both B strains and the C4.2 strain contain the prophage phi297. A homolog of this phage was described to be involved in enhanced biofilm formation by the expression of the bacteriophage control infection (*bci)* gene which is involved in the regulation of the quorum sensing system, motility as well as biofilm and pyocyanin production [42]. In all phi297 prophage regions found in the WM isolate genes, the *bci* gene could be identified with a query coverage of 95% or more. Investigations at the protein level revealed that most mutations do not interfere with the protein sequence (silent mutations). Nonetheless, the C4.2 strain revealed a mismatch of ten bases including the stop codon. A striking difference is the presence of the prophage Dobby which is only present in the strong biofilm producers B1.2 and B2.1. The ϕCTX-like prophage Dobby was first described in *P. aeruginosa* isolates from kidney stone microbiota [90]. To the best of our knowledge, there are currently no publications linking prophage Dobby with biofilm formation. An additional reason for the weak biofilm production of the C4.2 strain might be an activation of lytic phages during cultivation. It is also possible that both C strains preferably form flocs, biofilms without any surface adherence [91], or pellicles at the air–liquid interface [92] and are therefore not detectable as strong biofilm producers in the assays used here. Strong pellicle formation at the air–liquid interface could be observed for the C1.3 strain (Figure 1A). 

Another important virulence factor of *P. aeruginosa* is the outer membrane lipopolysaccharide (LPS) layer. LPS has been associated with antibiotic resistance and the vulnerability of *P. aeruginosa* against bacteriophages [93,94,95]. During biofilm formation, the LPS structure is highly dynamic and plays an important role during the initial attachment [96]. One component of the LPS layer is the O antigen, consisting of a common polysaccharide antigen (CPA) and the O-specific antigen (OSA). For the establishment of a mature biofilm, CPA has been reported to play a more crucial role than OSA [94]. The OSA is the most variable region of LPS and defines the *P. aeruginosa* serotype. The most common serotyping scheme is the International Antigenic Typing Scheme (IATS) which classifies *P. aeruginosa* strains into one of 20 O serotypes [97,98,99]. Within these 20 serotypes, 11 unique OSA gene clusters occur [100]. In the healthcare context, the knowledge of the *P. aeruginosa* serotype is crucial for monitoring outbreaks [68]. Specific serotypes occur more often in the clinic and are more commonly associated with multidrug resistance (MDR) [101,102,103]. According to clinical data, serotypes O1, O6, O11 and O12 account for up to 70% of infections [104,105,106,107]. Serotyping of the household appliance isolates revealed that three isolates belong to these serotypes frequently associated with infections (B1.2 and B2.1 = O6, C1.3 = O1).

Additionally, other virulence factors and biofilm-associated genes linked with adherence, mobility, antimicrobial activity/competitive advantages, alginate, quorum sensing, effector delivery system, exoenzymes, exotoxins, immune modulation, nutritional metabolic factors and regulation were investigated. From a total of 285 genes, a variation in 42 genes could be identified. Variation within the gene clusters associated with the siderophore pyoverdine (a nutritional/metabolic factor), secreted effectors, quorum sensing, LPS, flagella and Type IV pili (mobility) could be found. The siderophore pyoverdine enables the acquisition of Fe(III) ions but has also been reported as being essential for the virulence of *P. aeruginosa* [108,109,110]. The *pvdE* gene, a precursor for pyoverdine synthesis, is only present in the C1.3 isolate. The synthesis of pyoverdine requires a special σ factor, *PvdS*, which is deleted in all isolates except C1.3, and this might give an explanation for the yellow-greenish color, which is only observed for C1.3. Secreted effector proteins (toxins) related to the Type III secretion system (*exoS*, *exoT*, *exoU*, *exoY*) play a crucial role during infection [34,35,36,37]. The secreted effector *exoU* was frequently found in *P. aeruginosa* strains isolated from human keratitis infections [111,112,113]. In contrast, the effector *exoS* was primarily found in strains isolated from cystic fibrosis patients [111,114]. *ExoU* was present only in the C1.3 strain, while the strain has a lack of exoS. Furthermore, the *pldA* gene was found in C1.3. It encodes for *phospholipase* D, a Type VI SS secreted effector, which promotes chronic infections [115,116]. Crucial systems for proper biofilm formation are the separate but interrelated *las*, *rhl* and *mvfR* (also known as *PqsR)* and IqsR quorum sensing systems [28,73,117]. They regulate a broad range of virulence factors of *P. aeruginosa*. The *P. aeruginosa* strain PAO1 with a defective *las* quorum sensing product has been reported to form EPS but was unable to build significant biofilm structures [118]. The *las* system (*lasI* and *lasR*) is absent in the C1.3 strain, which might explain the observed weak biofilm production. Additionally, a subset of flagellar genes including *FliD* and type IV pili genes were absent. During initial biofilm formation, flagellar genes such as the flagellar cap protein fliD together with flagellin are involved in adherence [119,120]. The role of flagella has also been reported to play a crucial role in infections [121]. During infection, flagella can elicit a strong NFkB-mediated activation of the host immune response via Toll-like receptor 5 (TLR5) and a caspase-1-mediated response via the Nod-like receptor Ipaf [122]. However, *P. aeruginosa* strains lacking some of the flagellar genes have been isolated from infectious patients, revealing that infection is still possible [111]. Since many virulence factors such as flagella, pili, the secretion systems and quorum sensing systems are directly linked to biofilm formation and thus to severe infections, a study of these factors is of great interest. Interestingly, a recent study, by Subedi et al., investigated the prevalence of 147 virulence genes of 22 clinical *P. aeruginosa* isolates from microbial keratitis and from CF patients [111]. They also observed a similar prevalence of genes encoding for flagella, exotoxins, phospholipase D and pyoverdine as observed in this study. This suggests that the genomic variation observed for environmental strains as investigated here may influence the infection site. 

Taken together, the knowledge about the genetic heterogeneity of *P. aeruginosa* biofilms in household appliances improves the knowledge of the household being a potential reservoir of clinical (outbreak) strains. High concentrations of detergents as well as flushing stress may exert strong adaptation pressure on *P. aeruginosa* in household appliances such as WMs. Enhanced biofilm formation could be a mechanism to cope with these stressors. Therefore, future studies on the variation of household acquired opportunistic pathogens are needed to provide a better understanding about selection pressure within household appliances. 

## 5. Conclusions

This study investigated the phenotypic and genetic variations of the opportunistic pathogen *P. aeruginosa* isolated from different parts of household washing machines. Phenotypic analysis revealed that the strains are highly diverse in their biofilm-forming potential and also regarding their growth characteristics. Genetic investigations emphasize the relevance of an intact quorum sensing system. Furthermore, different prophages could be related to biofilm formation. Here, the prophage Dobby was only detected in the strong sessile biofilm-forming strains B1.2 and B2.1.

## Figures and Tables

**Figure 1 microorganisms-10-02508-f001:**
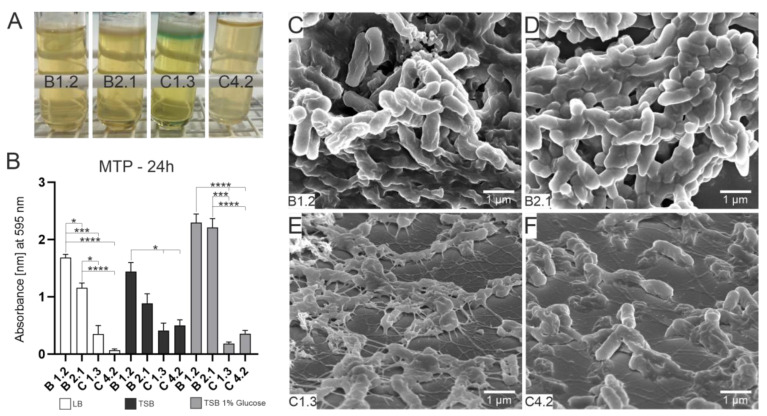
Phenotyping of the four *P. aeruginosa* WM isolates. Static growth behavior was investigated in LB medium for 72 h (**A**). Only C1.3 was able to produce a yellow-greenish color at the water–air interface. Microtiter plate assay using crystal violet staining was performed for a quantification of the biofilm-forming potential of the four isolates (**B**). The two B isolates had significantly higher absorbance values, indicating them to be the strongest biofilm producers. Means ± standard error of the mean (SEM) were statistically analyzed by Brown–Forsythe and Welch ANOVA tests (*p* < 0.05 (*), *p* < 0.001 (***), *p* < 0.0001 (****)). Data passed the D’Agostino and Pearson normality test (alpha = 0.05). Ultrastructural analysis of biofilm formation was performed by SEM (C-F). SEM images reveal that both B isolates show vertical growth and even biofilm-typical structures such as supply channels and matrix production (**C**,**D**). For the C isolates, initial biofilm stages can be observed (**E**,**F**), such as network structures and initial matrix production, but no multilayered growth appeared.

**Figure 2 microorganisms-10-02508-f002:**
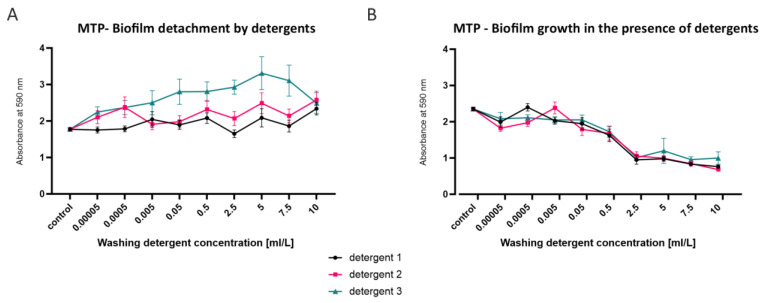
Biofilm detachment and biofilm formation of WM isolate B2.1 with liquid washing detergents. The microtiter plate assay followed by crystal violet staining was performed for quantification of biofilm detachment by liquid washing detergents (**A**). Biofilm formation in the presence of liquid detergents was additionally tested (**B**). Means ± standard error of the mean (SEM) are plotted.

**Figure 3 microorganisms-10-02508-f003:**
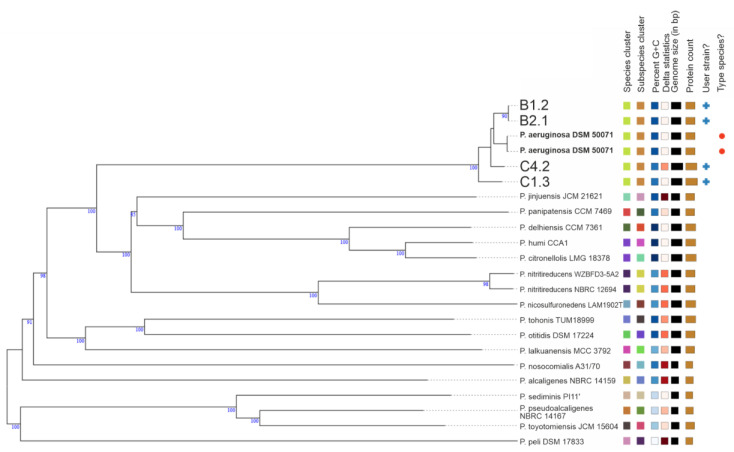
Genome-scale GBDP gene sequence tree. Whole genome sequences were used for additional taxonomic classification of the strains. Phylogenetic genome-scale GBDP tree and 16S rRNA gene sequence tree (Appendix A) reveal that the four WM isolates belong to the species *P. aeruginosa*. Generated by using the Type (Strain) Genome Server (TYGS) [56,57].

**Figure 4 microorganisms-10-02508-f004:**
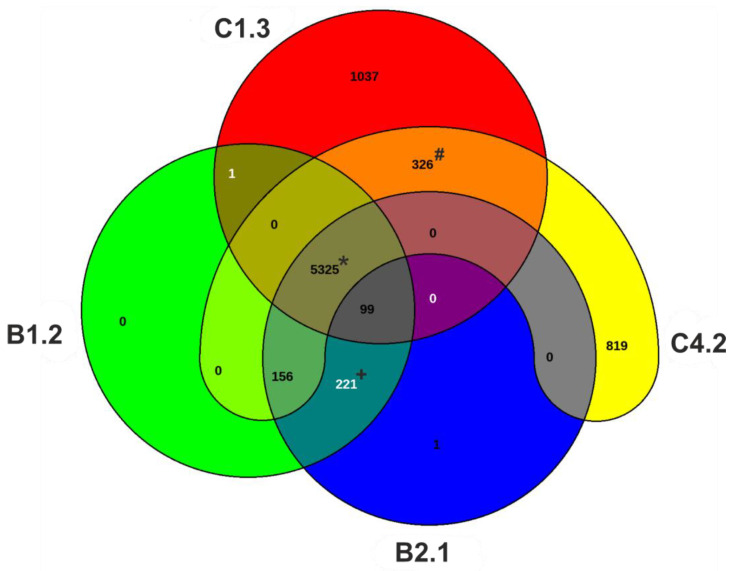
Venn diagram displaying the pan-genome of the *P. aeruginosa* isolates. The four isolates have a core genome with 5325 CDS (asterisks). The B isolates shared 221 orthologous CDS (cross). Orthologous shared CDS among the C isolates are 326 genes (hashtag). Venn diagram was produced using EDGAR 3.0.

**Figure 5 microorganisms-10-02508-f005:**
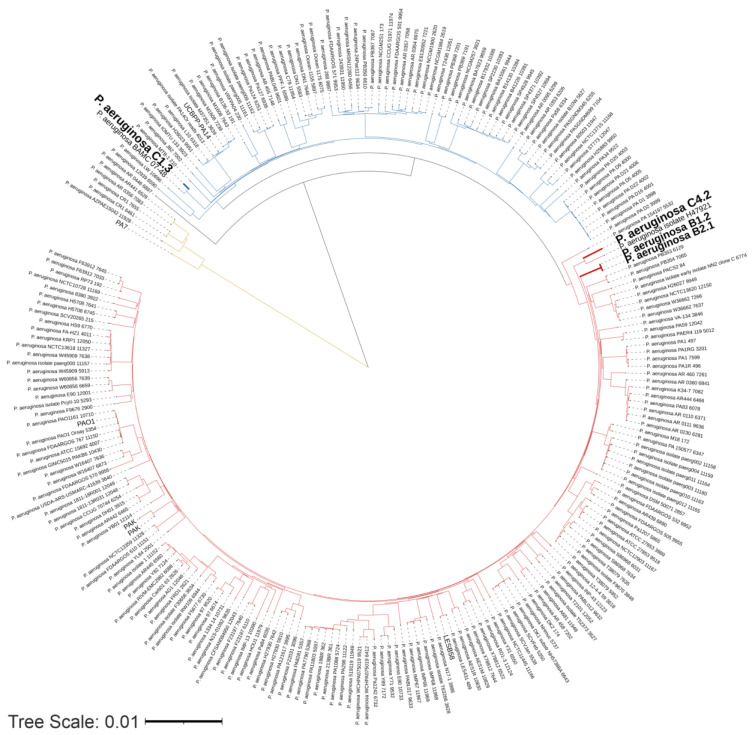
Phylogenetic analysis of *P. aeruginosa* WM isolates. Circular tree built with the four WM isolates and 233 *P. aeruginosa* complete genome sequences taken from the Pseudomonas Database, including popular/reference strains PAO1, LESB48, UCBPP-PA14 and PA7 [58,59]. Three distinct clusters are visible and additionally colored in yellow, blue and red.

**Figure 6 microorganisms-10-02508-f006:**
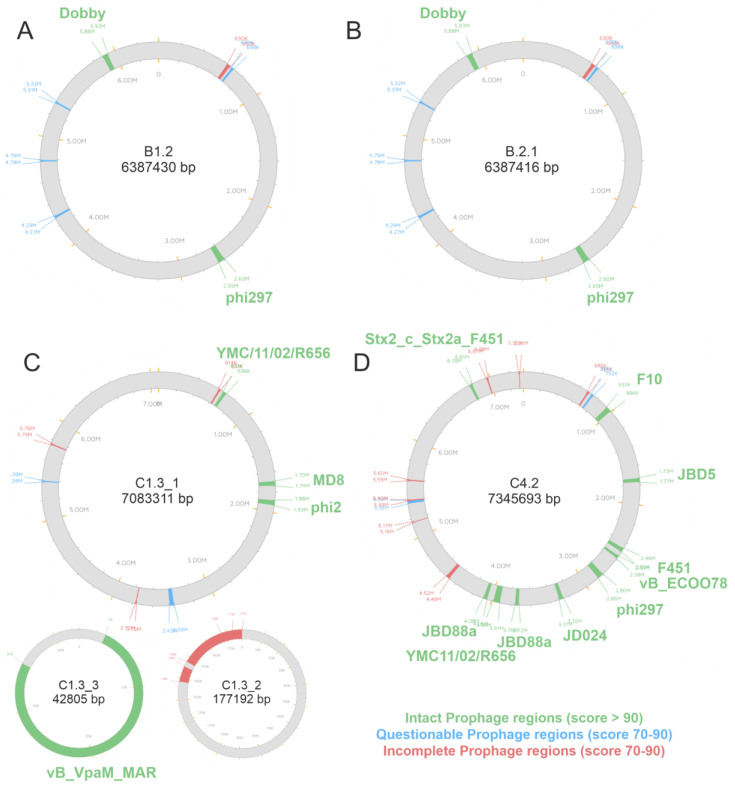
Identification of prophage regions. Prophages were identified using the PHASTER web tool and are indicated at their integration sites. Intact prophage regions are displayed in green, questionable prophage regions in blue and incomplete prophage regions in red. Only intact prophage regions are indicated.

**Figure 7 microorganisms-10-02508-f007:**
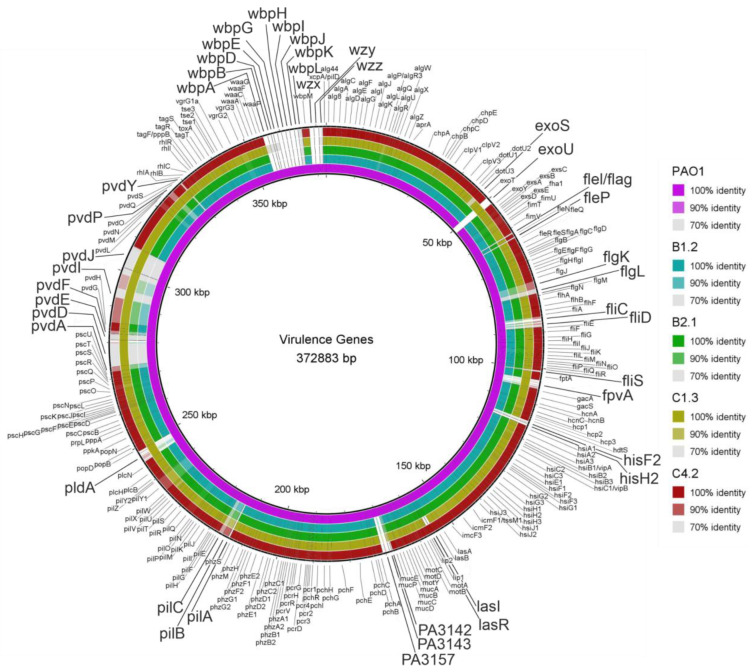
A circular illustration of the comparison of 285 virulence- and biofilm-associated genes among the WM isolates of *P. aeruginosa*. BLAST Ring Image Generator (BRIG) was used to create the circular map. All genes were taken from the reference strain PAO1, except for *exoU* which was taken from the reference strain PA14. Genes with identity scores less than 90% have bigger labels.

**Table 1 microorganisms-10-02508-t001:** Sampling location and name of *P. aeruginosa* WM isolates. The capital letters in the isolate names indicate the corresponding washing machine origin.

WM Compartment	Name of WM Isolates
detergent compartment	B1.2
detergent enema	B2.1
detergent compartment	C1.3
sealing rubber	C4.2

**Table 2 microorganisms-10-02508-t002:** Table of *P. aeruginosa* WM isolates and *P. aeruginosa* reference strains. Sequence types were determined using the MLST, by scanning the isolates/reference strains against PubMLST typing schemes. In silico serotyping was performed using PAst, based on BLAST analysis of the O-specific antigen (OSA) gene cluster. Information taken from the Pseudomonas Genome Database is marked with a star (*) [58,59].

Sampling Site	Isolate	Length (bp)	GC (%)	Total Genes (CDS)	Sequence Type	Serotype Group
**WM isolates**						
detergent compartment	B1.2	6,387,430	66	5930 (5851)	369	O6
detergent enema	B2.1	6,387,416	66	5930 (5851)	369	O6
detergent compartment	C1.3	7,303,308	65	7034 (6952)	313	O1
sealing rubber	C4.2	7,345,693	66	6987 (6900)	2844	O13
**Reference strains**						
human wound	PAO1	6,264,404 *	67 *	5708 (5587) *	549	O5
human burn wound	UCBPP-PA14	6,537,648 *	66 *	5983 (5894) *	253	O10
human non-respiratory clinical isolate	PA7	6,588,339 *	66 *	6369 (6286) *	1195	O12
human sputum	LESB58	6,601,757 *	66 *	6028 (5927) *	146	O6

## Data Availability

All raw sequence data as well as the assembled genomes are available via the BioProjects PRJNA838003 (B1.2), PRJNA838002 (B2.1), PRJNA838005 (C1.3) and PRJNA838004 (C4.2), respectively. All other presented data generated by our experiments are included in the manuscript and Appendix A.

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
