# Peer review of "Genome-Based Analysis of Virulence Factors and Biofilm Formation in Novel P. aeruginosa Strains Isolated from Household Appliances"

_microorganisms, 2022, doi:10.3390/microorganisms10122508_

Round 1
Reviewer 1 Report
Kiel et al. describe opportunistic growth of Pseudomona aeruginosa and the associated biofilm production in the light of growth patterns, genetics, and prophage infiltration.
The experiments are well designed, the result figures are exceptionally well presented, and the experimental data is embedded logically in the context of the knowledge of the scientific community referenced.
Reviewer 2 Report
The authors sequenced four Pseudomonas aeruginosa genomes and did genome analysis to explore the isolates’ ability in biofilm formation and virulence. Their results are important to understand why bacteria can adapt in wash machine (harsh environment) and survive in an environment with high concentrations of detergents, which provides greater insights into pathogenicity and underlying mechanisms. In addition, this study explored the possibility of these isolates become to pathogens in the society.
In general, this study is well-performed, but I do have some minor comments.
The introduction section can be improved to cover the knowledge gap, hypothesis, and then your goal.
This study discussed a lot about the virulence which does not match the title or the connection between the virulence and the biofilm formation is not well-discussed in this paper.
Please see my other comments below:
Line 82: could you provide more information about the mechanism? knowledge gap? And then what you want to address in your study?
Line 110: be more specific about the culture condition. for example, agitation rate? with or without light? what's the rationale that growth behavior experiment was performed without agitation?
Line 121: could you give more detail about the cell density (i.e., cells/mL) when OD600 = 0.01 for your strains?
Line 127: just curious, wash with water instead of buffer solution (i.e., PBS), will this affect the feature of biofilm?
Line 196: please provide more details about sequence data analysis. for example, quality control, any filter criteria, assembly parameters setup, reference database, etc.
Line 228: could you also provide more details about the BLAST parameters you used to do search the genes?
Line 263: any explanation?
Line 281: could you also add statistical analysis method details in your method material section? What software and version you used?
Line 331: any statistical methdo was used to support you to say it's significantly larger? if no statistics was used, please do not use the word "significant".
although the genome size is larger, what about the number of genes? or the number of genes encoding biofilm formation?
Line 346: any explanation about this difference?
Line 386: what's the rationale of this hypothesis since incomplete assembly could be attributed to many reasons?
Line 393: any previous studies from your team of any other team to support that prophages at the same location indicates that the different isolates are the same strain?
Line 518: could you explain more how is this related to biofilm formation?
Line 566: either revise your title a little bit to also cover virulence or revise your discussion to focus more on biofilm formation
